# Ribo-On and Ribo-Off tools using a self-cleaving ribozyme allow manipulation of endogenous gene expression in *C. elegans*

Jie Fang[1,2,3,7], Jie Wang[1,2,7], Yuzhi Wang[1,2], Xiaofan Liu[1,2], Baohui Chen [3,4,5,6✉] & Wei Zou [1,2✉]

Investigating gene function relies on the efficient manipulation of endogenous gene expression. Currently, a limited number of tools are available to robustly manipulate endogenous gene expression between "on" and "off" states. In this study, we insert a 63 bp coding sequence of T3H38 ribozyme into the 3' untranslated region (UTR) of *C. elegans* endogenous genes using the CRISPR/Cas9 technology, which reduces the endogenous gene expression to a nearly undetectable level and generated loss-of-function phenotypes similar to that of the genetic null animals. To achieve conditional knockout, a cassette of *loxP*-flanked transcriptional termination signal and ribozyme is inserted into the 3' UTR of endogenous genes, which eliminates gene expression spatially or temporally via the controllable expression of the Cre recombinase. Conditional endogenous gene turn-on can be achieved by either injecting morpholino, which blocks the ribozyme self-cleavage activity or using the Cre recombinase to remove the *loxP*-flanked ribozyme. Together, our results demonstrate that these ribozyme-based tools can efficiently manipulate endogenous gene expression both in space and time and expand the toolkit for studying the functions of endogenous genes.

[1] The Fourth Affiliated Hospital, Zhejiang University School of Medicine, 322000 Yiwu, China. [2] Institute of Translational Medicine, Zhejiang University, 310058 Hangzhou, China. [3] Department of Cell Biology and Bone Marrow Transplantation Center of the First Affiliated Hospital, Zhejiang University School of Medicine, 310058 Hangzhou, China. [4] Zhejiang Laboratory for Systems & Precision Medicine, Zhejiang University Medical Center, Hangzhou, China. [5] Institute of Hematology, Zhejiang University & Zhejiang Engineering Laboratory for Stem Cell and Immunotherapy, Hangzhou, China. [6] Zhejiang Provincial Key Laboratory of Genetic & Developmental Disorders, Hangzhou, China. [7] These authors contributed equally: Jie Fang, Jie Wang. ✉email: baohuichen@zju.edu.cn; zouwei@zju.edu.cn

In the post-genomic era, one of the most challenging things is to assess gene functions in development and disease. To achieve this, one must disrupt a given gene and examine the loss-of-function phenotype. At the DNA level, gene-targeted mutagenesis mediated the CRISPR/Cas9 has been proven to be the most efficient method[1–3]. At the RNA level, RNA interference and recently developed CRISPR interference technologies can be conveniently used in a high throughput manner[4,5]. However, potential off-target effects should be carefully considered when using these methods. At the protein level, methods such as degron-induced protein degradation show robust efficacy for a large portion of proteins[6]. However, some of the proteins are difficult to be degraded due to their specific subcellular localization or high stability. Thus, additional efficient and specific methods for gene regulation are still needed.

Ribozymes are catalytic RNA molecules capable of RNA cleavage, which act both in trans and in cis[7–9]. Trans-acting ribozymes use their base-pairing region, or arms, to specifically bind with and cut the target RNAs. However, the overall efficiency is not high. Recently, cis-acting ribozymes, such as a class of type III hammerhead ribozymes, are highly efficient in mediating the inactivation of gene expression due to their self-cleavage-induced mRNA decay when inserted into targeted RNA molecules[9,10]. T3H38, one of the hammerhead ribozymes, inhibited transgene expression by ~730-fold compared to the catalytically inactive form of T3H38. The transgene expression could be partially restored by a steric-blocking antisense oligonucleotide, which binds to and inactivates the self-cleavage of T3H38[10]. In C. elegans, a tetracycline-dependent ribozyme was designed to mediate conditional expression of exogenous transgenes in vivo. However, the transgene reporters controlled by this riboswitch exhibited strongly leaky expression when tetracycline was absent. Additionally, the expression was increased with narrow dynamic ranges (only by 2-fold)[11]. In another study, the glmS ribozyme was inserted into the C-terminus of endogenous PfDHFR-TS locus in the model parasite Leishmania tarentolae. Glucosamine treatment activated the self-cleavage activity of glmS ribozyme, leading to the efficient knockdown of the PfDHFR-TS gene[12]. However, whether ribozymes can serve as efficient and specific tools to manipulate the expression of endogenous genes in organisms besides Leishmania tarentolae has not been explored.

In this study, we tested the self-cleavage efficiency of the T3H38 ribozyme in the multi-cellular model organism C. elegans. Inserting T3H38 into the 3' UTR of endogenous protein-coding genes efficiently eliminates RNA and thus represses protein expression. Injecting antisense oligonucleotide successfully restored gene expression and rescued the loss-of-function phenotypes. Moreover, we combined the ribozyme and the Cre/loxP system to create Ribo-On and Ribo-Off tools, which can achieve spatiotemporal manipulation of the endogenous genes of interest. We anticipate these new methods may work in other multi-cellular organisms, including mammals, and will expand the toolbox of gene regulation.

## Results

### T3H38 ribozyme efficiently cis-cleaves endogenous mRNAs in C. elegans.

To test whether the T3H38 ribozyme is adaptable for C. elegans, we envisioned that CRISPR/Cas9-mediated knock-in of the active T3H38 ribozyme into the 3' untranslated region (UTR) of an endogenous gene would lead to ribozyme-mediated self-cleavage and mRNA degradation as the mRNA does not contain a 3' UTR[13]. In contrast, the use of a catalytically inactive mutant form would generate a stabilized mRNA and subsequently guide normal protein production (Fig. 1a). Toward this goal, we inserted the coding sequence for the green fluorescence protein (gfp), gfp-inactive T3H38 (gfp-Rz*), or gfp-active T3H38

(gfp-Rz) into the immediately upstream of the 3' UTR of three endogenous genes, respectively (Supplementary Fig. S1)[10]. These genes are argn-1 (a gene required for normal mitochondrial morphology), scav-3 (a gene required for normal lysosomal morphology) and sax-7 (a gene required for proper dendrite branching and guidance) (Fig. 1b)[14–17]. In the three strains inserted with C-terminal gfp, the morphologies of mitochondria, lysosomes or the PVD dendrites were similar to that of wild-type control, suggesting that tagging these genes with a C-terminal gfp does not affect their functions (Fig. 1c, d). Next, we examined the three strains in which the target genes were tagged with gfp-inactive T3H38. We found that 97%, 83% and 89% animals showed normal morphologies of mitochondria, lysosomes, and dendrites, respectively (Fig. 1c, d). Notably, quantitative fluorescent imaging revealed that the intensity of GFP-Rz*-fused ARGN-1 and SCAV-3 proteins were downregulated by 79% and 64%, respectively, while the level of GFP-Rz*-fused SAX-7 protein was only slightly reduced (Fig. 1e). These results suggest that placing the inactive ribozyme in the 3' UTR may affect the expression of some target genes, presumably due to various factors such as defects in mRNA stability, post-transcriptional modification, transport or translation. However, the remaining proteins of ARGN-1 and SCAV-3 appeared to be sufficient to maintain the normal morphology of organelles. We then examined the three strains carrying gfp-active T3H38 insertions. As expected, they all exhibited phenotypes similar to strong loss-of-function or null mutants[14–17], suggesting that the endogenous gene expression was almost completely suppressed, which is supported by our quantification of the GFP intensity (Fig. 1d, e). Quantitative reverse transcription PCR (qRT-PCR) analyses were consistent with the notion that active T3H38 efficiently cleaved target mRNAs in all strains (Fig. 1f).

To further demonstrate the efficacy and versatility of ribozyme-mediated gene regulation, we extended our investigation to include a broader range of genes. Specifically, we inserted loxP-T3H38-loxP into five additional genes, including dhgd-1, hphd-1, drp-1, mnr-1 and eff-1, which are essential for maintaining normal mitochondrial morphology or PVD dendrite arborization[16,18–21]. Our T3H38 ribozyme-based strategy achieved robust knockdown of endogenous protein expression for all these genes, as evidenced by the null-like loss-of-function phenotypes displayed by the T3H38 knock-in alleles (Supplementary Fig. S2a–c).

The above eight genes act in the epidermis to control proper morphologies of mitochondria and lysosomes within the epidermis and PVD dendrites[14–21]. To test whether the T3H38 ribozyme is effective in other types of tissues, we inserted the sequence of T3H38 into the endogenous locus of unc-86, mec-3, dma-1, hpo-30, kpc-1, rab-10 and lect-2, which function in the PVD neurons (for the first six genes) and body wall muscles (for lect-2), respectively, to control dendrite branching[22–29]. Our results indicate that all ribozyme knock-in strains, except kpc-1-loxP-ribozyme-loxP, showed almost identical phenotypes as the genetic null mutant animals, suggesting that T3H38 successfully blocks the endogenous expression of these six genes in neurons and muscles, respectively (Supplementary Fig. S2d, e). However, inserting the ribozyme into the kpc-1 locus did not result in a loss-of-function phenotype, suggesting that the self-cleavage activity might be affected when T3H38 was inserted into some specific rare sequence context. Together, these results demonstrate that T3H38 ribozyme efficiently eliminates endogenous gene expression in multiple types of tissues when inserted into the 5' of the 3' UTR for most of the genes we tested in C. elegans. We named this ribozyme-mediated inactivation of endogenous gene expression as Ribo-Off. Thus, robust gene knockout can be achieved with a single genetic modification by inserting the 63 bp T3H38 ribozyme sequence into the 3' UTR of a gene of interest.

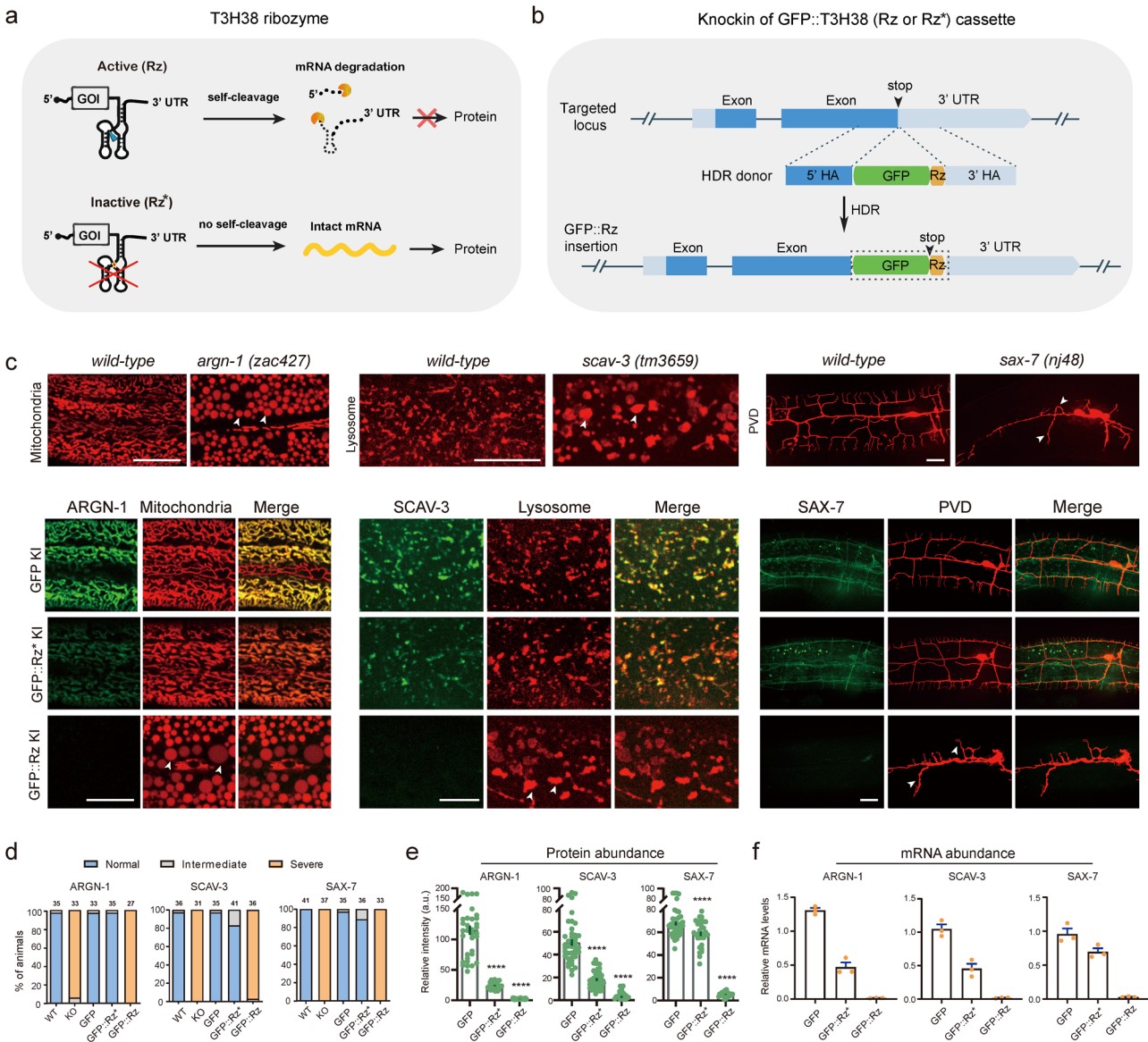

**Fig. 1 Knock-in of a ribozyme generates null-like mutants. a** Schematic of active T3H38 ribozyme (Rz) for inducing an off-switch of gene expression. Adapted with permission from Wurmthaler et al.[11]. An inactive T3H38 (Rz*) was generated as a control. **b** Schematic diagram of CRISPR/Cas9-mediated homologous recombination to integrate GFP, GFP::Rz or GFP::Rz* at the C-terminal of endogenous loci. **c** Confocal fluorescent images to show the morphology of mitochondria (*Pcol-19::mito-mKate*), lysosomes (*Pced-1::nuc-1::mCherry*) and PVD (*ser2prom::myr-mCherry*) in related to the protein expression level of ARGN-1::GFP, SCAV-3::GFP and SAX-7::GFP, respectively, under different conditions. 3-day-old adults were imaged for the *argn-1* and *scav-3* groups; 2-day-old adults were imaged for the *sax-7* group. Scale bars: 20 μm. Arrowheads: enlarged mitochondria/lysosome or dendrite branching defects. *argn-1(zac427)*, *scav-3(tm3659)* and *sax-7(nj48)* are deletion mutants, resulting in strong loss-of-function/null alleles, and are used for comparisons. **d** Quantifications of the proportion of animals containing normal, intermediately defective and severely defective morphology of mitochondria (*argn-1* group), lysosomes (*scav-3* group) and PVD dendrites (*sax-7* group), respectively. The number of animals quantified for each group was indicated above the columns. **e** Expression of ARGN-1::GFP, SCAV-3::GFP and SAX-7::GFP, as shown in (**c**), were quantified based on fluorescent imaging. Data are displayed as mean ± s.e.m. Each dot represents a single worm. Number of animals quantified: *n* = 35 for ARGN-1::GFP; *n* = 37 for ARGN-1::GFP::Rz*; *n* = 31 for ARGN-1::GFP::Rz; *n* = 41 for SCAV-3::GFP; *n* = 40 for SCAV-3::GFP::Rz*; *n* = 34 for SCAV-3::GFP::Rz; *n* = 41 for SAX-7::GFP; *n* = 32 for SAX-7::GFP::Rz*; *n* = 36 for SAX-7::GFP::Rz. ****$p$ < 0.0001 (one-way ANOVA with the Tukey correction). **f** Relative mRNA abundance of target genes at different conditions in (**c**) measured by quantitative RT-PCR. Three biological replicates were quantified and shown. All values are displayed as mean ± s.e.m. Source data are provided as a Source Data file.

We next compared the knockdown efficiency of Ribo-Off and RNA interference for six genes based on the severity of their loss-of-function phenotypes. Our results showed that inserting the ribozyme into *argn-1*, *drp-1* and *scav-3* led to almost 100% of animals displaying severely enlarged mitochondria or lysosome phenotypes, whereas gene-specific dsRNAs resulted in intermediate phenotypes in small but significant proportions of

animals (Fig. 2a, b). Loss of endogenous *eff-1* caused ectopic branching in the lateral regions, while loss of *dma-1* or *sax-7* resulted in reduced higher-order branch formation (tertiary and quaternary branches)[16,20,22]. The whole-body Ribo-Off alleles of *eff-1*, *dma-1* and *sax-7* phenocopied the previously reported strong loss of function or null alleles, while *eff-1* (RNAi), *dma-1*(RNAi) and *sax-7* (RNAi) animals showed less severe dendrite

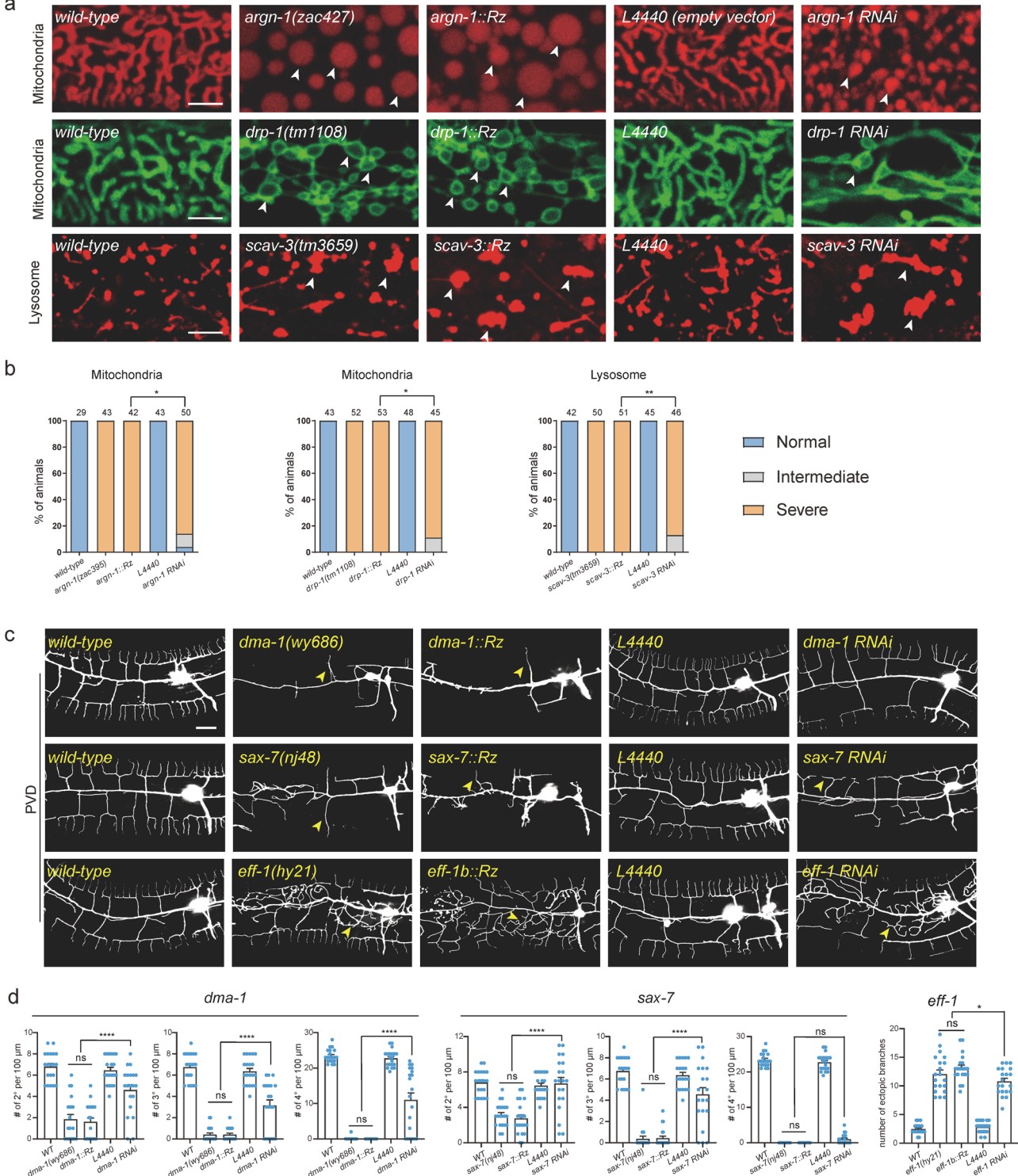

**Fig. 2 A comparison of ribozyme-mediated gene inactivation with RNA interference. a** Confocal images of *C. elegans* expressing transgenic markers to label mitochondria or lysosomes in wild-type and mutant animals. Mitochondria were labeled via *zjuSi47[Pcol-19>mito::mKate2]* or *zjuSi48[Pcol-19>tomm-20::gfp]*. Lysosomes were labeled via *zacSi13(Psemo-1>ctns-1-mcherry)*. Arrowheads: abnormally enlarged mitochondria or lysosomes. Scale bars: 5 μm. **b** Quantification of organelle morphology phenotypes. *$p < 0.05$; **$p < 0.01$, $\chi2$ test. The number of animals quantified for each group was indicated above the columns. **c** Confocal images showing dendrite morphologies in wild-type and mutant animals. PVD dendrites were labeled via *wyIs592[PVD>myr-gfp]*. Scale bars: 20 μm. **d** Quantifications of dendrite branching phenotypes in wild-type and mutant animals. All values are presented as mean ± s.e.m. For *sax-7(nj48)* and *sax-7::Rz*: $n = 21$ animals; For all other genotypes: $n = 20$ animals. ns: not significant. *$p < 0.1$. ****$p < 0.0001$ (one-way ANOVA with the Tukey correction). Source data are provided as a Source Data file.

branching defects (Fig. 2c, d). Overall, our findings suggest that Ribo-Off outperformed RNAi, at least for the six genes we tested.

**Turn off endogenous gene expression in space and time via Ribo-Off.** Encouraged by the whole-body Ribo-Off described above, we sought to use ribozyme to perform conditional gene knockout. To achieve this purpose, we used CRISPR/Cas9 technology to insert *gfp::loxP::let-858* transcriptional termination sequence (TTS)::*loxP::T3H38* ribozyme immediately after the stop codons of the abovementioned three genes, *argn-1*, *scav-3* and *sax-7*[30,31]. Theoretically, the endogenous genes can use the *let-858* 3' UTR to stabilize the mRNAs without the expression of the recombinase Cre protein. However, when Cre is expressed in a specific tissue (via tissue-specific promoters) or at a specific developmental stage (via the heat-shock promoter), the *loxP- TTS-loxP* cassette will be deleted from the endogenous loci, and the mRNAs will be unstable and degraded due to T3H38-mediated self-cleavage, thus resulting in conditional knockout of target genes (Fig. 3a).

Indeed, all the strains carrying *gfp::loxP::TTS::loxP::T3H38* ribozyme knock-in were indistinguishable from the wild-type and the *gfp* knock-in control groups for the morphologies of mitochondria, lysosomes and dendrites, respectively (Fig. 3b and Supplementary Fig. S3a, b). Specific expression of Cre in the epidermis significantly reduced protein expression and generated defective morphologies, similar to that of the whole-body knockout animals, suggesting that the functions of these genes in the epidermis are dramatically impaired (Fig. 3b and Supplementary Fig. S3a, b). As a control, expressing the Cre in the muscle cells neither affected the expression of these genes nor generated any loss-of-function phenotypes in the epidermis, demonstrating that the conditional knockout is specific (Fig. 3b and Supplementary Fig. S3a, b). To test whether we could regulate the expression of endogenous genes in a temporal manner, we used a heat-shock promoter to drive Cre expression at specific time points. When the animals were not treated with heat-shock, mitochondrial or lysosomal morphology was normal. In contrast, heat shock for 1 h and recovery for 48 h almost totally blocked the expression of these genes and generated defective phenotypes similar to that of whole-body knockout control animals (Fig. 3c, d and Supplementary Fig. S3c–f).

**Ribo-Off can turn off essential genes in specific tissues.** Next, we sought to determine whether our ribozyme-based conditional knockout strategy applies to essential genes. We inserted the *loxP::TTS::loxP::ribozyme* sequence immediately after the stop codons of *bicd-1*, an essential gene required for proper dendrite morphogenesis (Fig. 4a). *bicd-1* null mutants are embryonic lethal[32]. Interestingly, we isolated *bicd-1(zac51)* from a forward genetic screen for dendrite morphogenesis abnormal mutants, which showed an ectopic branch formation defect similar to the *bicd-1 (RNAi)* animals[32]. The mutant animals were viable, possibly because *zac51* is a partial loss-of-function allele. This strain was used as a positive control. We found that the *bicd-1::loxP::TTS::loxP::ribozyme* genetically modified animals developed and generated progenies normally without Cre expression. A previous study has reported that *bicd-1* acts in PVD neurons to control dendrite branching[32]. However, using our newly developed ribozyme-based conditional Ribo-Off technique, we found that knockout *bicd-1* in the PVD cell lineage or the muscles did not generate a dendrite branching defect, while inactivation of *bicd-1* in epidermis phenocopied the loss-of-function phenotype, demonstrating a non-cell autonomous function of BICD-1 controlling dendrite morphogenesis (Fig. 4b, c). Thus, by combining the ribozyme and Cre/LoxP systems, we were able to inactivate the expression of the endogenous genes in both space and time.

**Turn on endogenous gene expression via morpholino-regulated T3H38.** After demonstrating that T3H38 ribozyme could be used to manipulate endogenous gene expression from "On" to "Off", we sought to determine whether it can act as an "Off-to-On" tool. Antisense oligonucleotide v-M8 induces a conformational change of T3H38, thereby blocking the self-cleavage activity of T3H38 ribozyme[10]. This system successfully induced the expression of an exogenous transgene in mice (Fig. 5a). To test this, we selected *argn-1* as the target, using strains carrying *gfp*, *gfp::Rz** or *gfp::Rz* insertion at *argn-1* locus (Fig. 5b, c). We then first tried whether injecting the v-M8 morpholino could regulate the expression of endogenous genes from "Off" to "On" state. We injected v-M8 into the *argn-1::gfp::Rz* animals on day 1 of adulthood and detected the protein expression after 48 h (Fig. 5d). Quantitative analysis of ARGN-1::GFP levels revealed that endogenous *argn-1* expression was partially recovered in a dose-dependent manner. Injection of morpholino at 250 μM successfully restored the gene expression to levels comparable to those of the inactive T3H38 control group, suggesting an efficient block of the ribozyme-mediated self-cleavage (Fig. 5e, f). We also examined the mitochondrial morphology in the animals injected with morpholino. Approximately 50% to 80% of animals showed wild-type-like normal mitochondrial morphology when injected with 50 to 250 μM morpholino. Moreover, 250 μM morpholino showed the best performance, which is consistent with the expression level (Fig. 5g). Notably, the injection of 400 μM morpholino was ineffective in turning on the expression of endogenous ARGN-1 or rescuing the abnormal mitochondrial morphology. The underlying mechanism behind this observation requires future investigation (Fig. 5e–g). Similar restoration of endogenous expression and protein function by injecting antisense morpholino was observed for *scav-3* and *sax-7* (Supplementary Figs. S4a–c and S5a, b).

To investigate the kinetics of the endogenous gene expression induced by morpholino, we conducted a time-course analysis of protein expression from the *argn-1::gfp::active T3H38* locus post morpholino injections. A mild induction was observed at 6 h. The expression reached the highest level at 48 h and then started to decline. At 120 h post-injection, the expression level was almost undetectable (Fig. 5h). We then analyzed mitochondria morphology to determine the functional outcome of endogenous gene expression. At 6 h post-injection, none of the animals showed normal mitochondrial morphology, suggesting that the mild expression level is insufficient for the ARGN-1's function. The percentage of animals showing normal mitochondrial morphology increased dramatically between 12 and 36 h and peaked at 48 h after injection, which was comparable to the inactive T3H38 control group (Fig. 5i). Similar induction was observed for endogenous *scav-3* by injection of morpholino (Supplementary Fig. S4a–c).

**Cre/loxP-mediated Ribo-On turns on gene expression in space and time.** The morpholino injection method can manipulate the expression of endogenous genes with temporal resolution but lacks spatial regulation. In addition, our findings suggest that the incorporation of an inactive ribozyme can perturb the expression of some endogenous genes we tested, possibly making it challenging to fully restore their expression levels to that of wild-type animals via morpholino injection. Therefore, inspired by the ribozyme-Cre-loxP-dependent conditional Ribo-Off strategy, we inserted the coding sequence for *gfp::loxP::Rz::loxP* immediately after the stop codons of endogenous genes. Theoretically, inserting the active ribozyme between the coding region and the 3' UTR would cause mRNA degradation and abolish the gene expression. Using the Cre recombinase, the expression could be restored in either a spatial or a temporal manner (Fig. 6a). To test

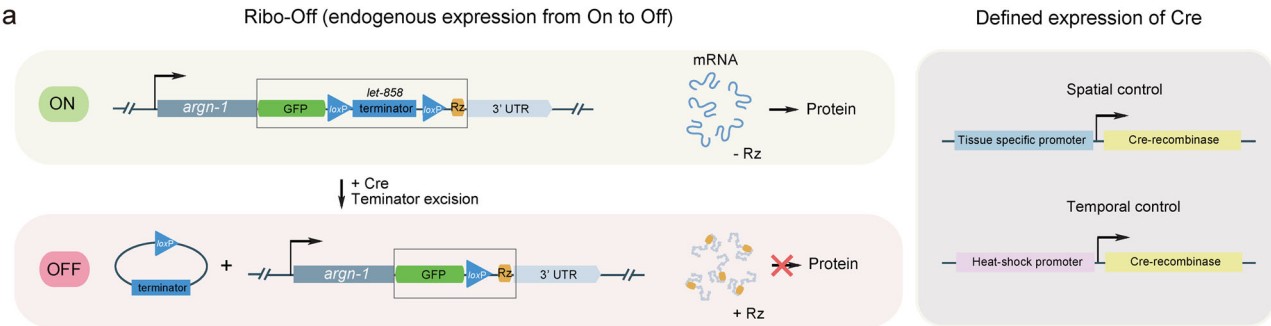

**Fig. 3 The Ribo-Off system enables spatiotemporal turn-off of endogenous gene expression. a** Schematics to show the designs for achieving spatiotemporal control of Cre expression (left) and the inducible T3H38 ribozyme/Cre-loxp system to conditionally knockdown endogenous expression. **b** Confocal fluorescent images to show the morphology of mitochondria labeled by MITO::mKate and ARGN-1::GFP expression levels under different conditions. 2-day-old adult animals were imaged. Scale bars: 5 μm. Arrowheads: enlarged mitochondria. **c** Relative expression of ARGN-1::GFP under various conditions as shown in (**b**). All values are presented as mean ± s.e.m. Each dot represents a single animal. n = 40 for ARGN-1::GFP; n = 33 for ARGN-1::loxP::TTS::loxP::Rz without Cre; n = 43 for ARGN-1::loxP::TTS::loxP::Rz with Skin Cre; n = 36 for ARGN-1::loxP::TTS::loxP::Rz with muscle Cre; n = 32 for ARGN-1::loxP::TTS::loxP::Rz with heat-shock Cre without heat-shock; n = 39 for ARGN-1::loxP::TTS::loxP::Rz with heat-shock Cre with heat-shock. ****p < 0.0001 (one-way ANOVA with the Tukey correction). **d** Quantifications of the proportion of animals containing normal, intermediately defective and severely defective morphology of mitochondria. The number of animals quantified for each group was indicated above each column. Source data are provided as a Source Data file.

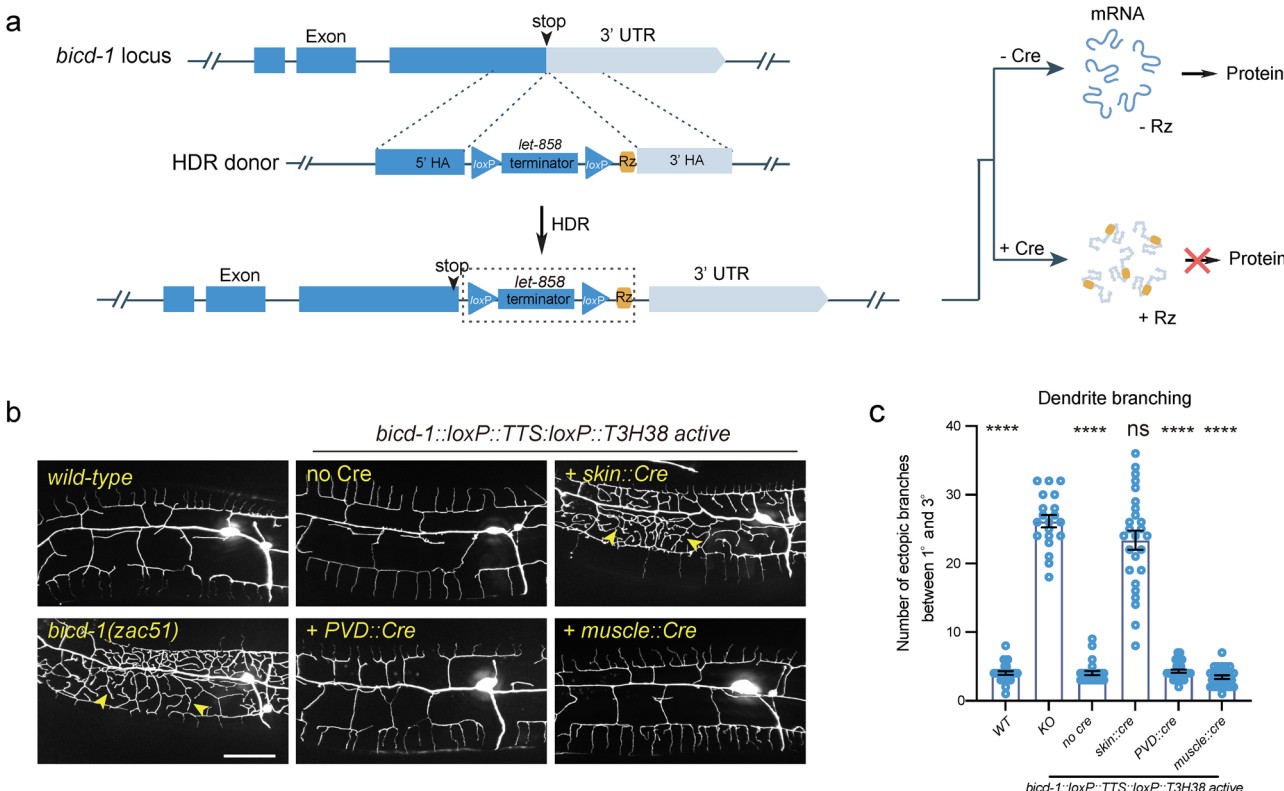

**Fig. 4 Conditional knockout of an essential gene using the Ribo-Off system. a** Schematic illustration of using CRISPR/Cas9-mediated homologous recombination to insert the GFP::loxP::TTS::loxP::Rz cassette to the C-terminal of the endogenous *bicd-1* locus, and the related outcome of protein expression with or without Cre expression. **b** Confocal fluorescent images to show the morphology of PVD dendrites. Cre recombinase was specifically expressed to delete the *let-858* transcriptional terminator in various tissues using different tissue-specific promoters. L4-stage animals with the heat-shock::Cre transgene were treated with heat-shock. 3-day-old adult worms were imaged for all groups. Scale bars: 20 μm. **c** Quantifications of number of ectopic dendritic branches between the primary dendrites and the tertiary dendrites in a region 100 μm anterior to the PVD cell body in different genetic backgrounds as shown in (**b**). All values are presented as mean ± s.e.m. Number of animals quantified: n = 27 for WT; n = 20 for *bicd-1(zac51)*; n = 26 for *bicd-1::loxP::TTS::loxP::Rz* without Cre; n = 26 for *bicd-1::loxP::TTS::loxP::Rz* with skin Cre; n = 28 for *bicd-1::loxP::TTS::loxP::Rz* with PVD Cre; n = 27 for *bicd-1::loxP::TTS::loxP::Rz* with muscle Cre. ns: not significant. ****$p < 0.0001$ (one-way ANOVA with the Tukey correction). Source data are provided as a Source Data file.

the idea, the *gfp::loxP::Rz::loxP* cassette was integrated immediately after the stop codon of *argn-1* gene, which did generate enlarged mitochondria similar to that of loss-of-function mutants. Skin-expressed Cre, but not the muscle-expressed Cre, restored both the mitochondria morphology and epidermal expression, suggesting that *argn-1* was successfully restored specifically in the epidermis. Using heat-shock-induced Cre, we were able to induce ARGN-1::GFP expression and rescue the *argn-1* loss-of-function phenotype in a temporal manner (Fig. 6b–d). Similar Off-to-On regulation was achieved for *dma-1*[22] (Supplementary Fig. S6a, b). We therefore named this ribozyme-dependent expression ON-switch as Ribo-On. Collectively, our results suggest that Ribo-On and Ribo-Off are effective tools for manipulating endogenous gene expression both in space and time, thus facilitating functional studies of specific genes in vivo.

## Discussion

Here we reported that ribozyme works efficiently to regulate endogenous gene expression in *C. elegans*, one of the most widely used model organisms. Knock-in of the 63 bp sequence of T3H38 ribozyme into the UTR region of the target gene generated strong loss-of-function whole-body mutants, which could be further restored by injecting antisense oligonucleotide. Furthermore, by combining the ribozyme and the Cre/loxP system, we could easily

manipulate endogenous gene expression between "On" and "Off" states both in space and time.

There are several advantages to applying Ribo-Off and Ribo-On systems for manipulating endogenous gene expression. First, the coding sequence of the T3H38 ribozyme sequence is only 63 bp, which is very short and easy to be inserted into a gene of interest using the CRISPR/Cas9 technology with a high success rate. In contrast, to generate a flanked-with-loxP (floxed) allele for "On" to "Off" control, it either requires precisely replacing a genomic DNA region that contains several important exons or inserting the two loxP sequences with two rounds of genome editing, which is usually time-consuming[33,34]. The loxP-TTS-loxP cassette was applied to manipulate endogenous gene expression from "Off" to "On". Compared to the floxed ribozyme, floxed TTS, such as the *let-858* terminator sequence (381 bp), is significantly longer, which may affect the CRISPR/Cas9-mediated knock-in efficiency. Second, manipulation of gene expression at the DNA level is usually not reversible. However, inactivated gene expression using Ribo-Off can be further turned on by either injecting antisense oligonucleotide or spatiotemporally controlled Cre expression, adding another layer of regulation. Third, both Ribo-On and Ribo-Off rely on CRISPR/Cas9-mediated knock-in, which ensures the specificity of gene regulation. RNA interference, CRISPR interference or the Cas13-mediated gene knockdown are effective tools for gene regulation at the mRNA

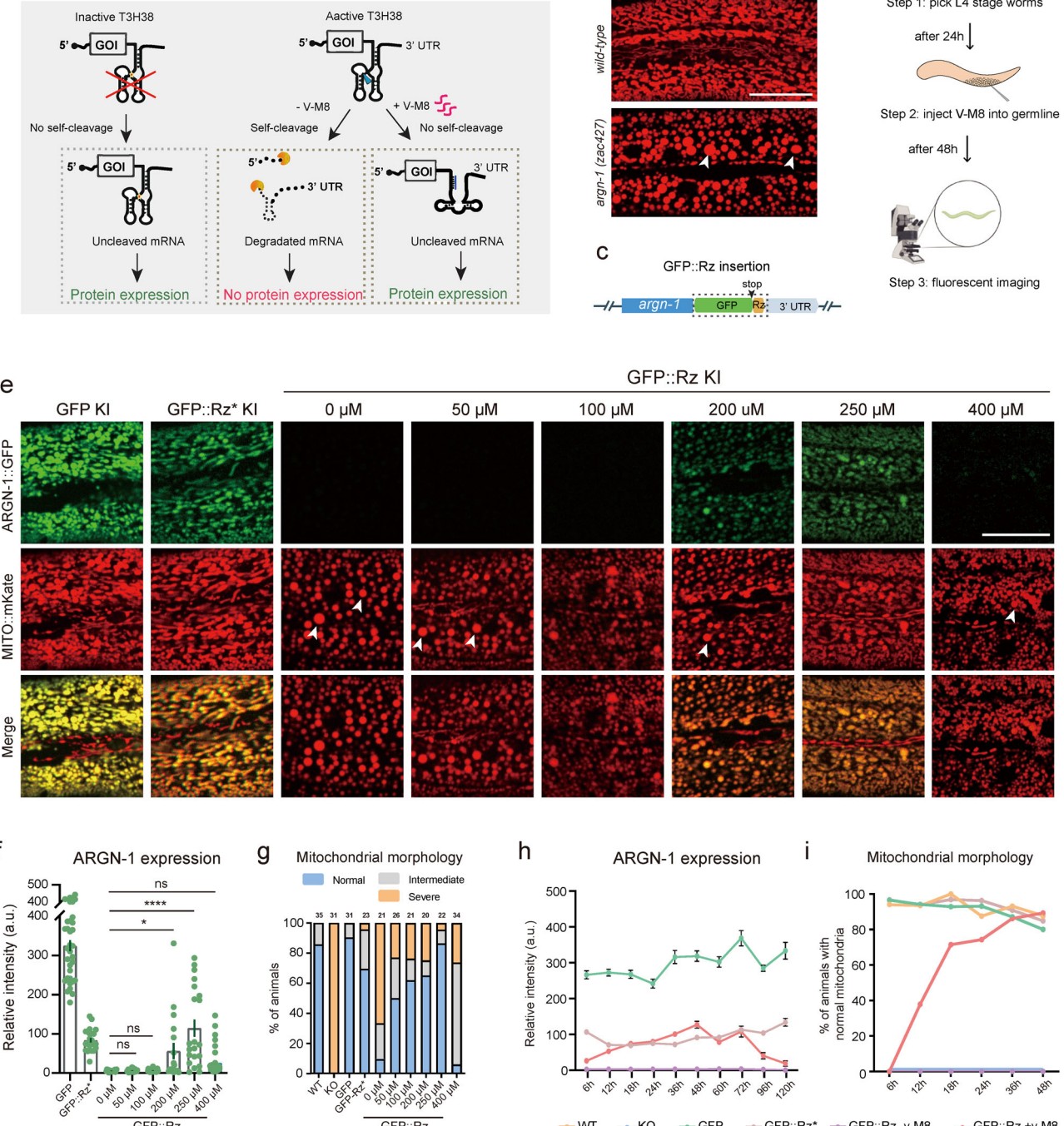

**Fig. 5 Temporal control of endogenous gene expression mediated by morpholinos. a** Diagram representing how an antisense oligonucleotide inactivates a ribozyme to restore protein expression and relative controls. Adapted with permission from Wurmthaler et al.[11]. **b** Confocal images to show the morphology of mitochondria in wild-type and mutant animals. Scale bars: 20 μm. **c** Diagram showing the cassette (outlined by a box) integrated into the C-terminal of *argn-1*. **d** A workflow of injecting morpholinos into the worm germ line and confocal imaging for quantifications. **e** Fluorescence images to show the morphology of mitochondrial and ARGN-1 expression under various conditions. Scale bar, 20 μm. **f** Relative protein expression of ARGN-1::GFP under different conditions as shown in (**e**). Each dot represents a single worm. Number of animals quantified: *n* = 35 for ARGN-1::GFP; *n* = 24 for ARGN-1::GFP::Rz*; *n* = 19 for ARGN-1::GFP::Rz without v-M8; *n* = 27 for ARGN-1::GFP::Rz injected with 50 μM v-M8; *n* = 21 for ARGN-1::GFP::Rz injected with 100 μM v-M8; *n* = 18 for ARGN-1::GFP::Rz injected with 200 μM v-M8; *n* = 21 for ARGN-1::GFP::Rz injected with 250 μM v-M8; *n* = 39 for ARGN-1::GFP::Rz injected with 400 μM v-M8. All data were shown as mean ± s.e.m. Ns: not significant. *$p < 0.05$. ****$p < 0.0001$ (one-way ANOVA with the Tukey correction). **g** Quantifications of mitochondria morphology under different conditions as shown in (**e**). The number of animals quantified for each group was indicated above each column. **h** The expression of ARGN-1::GFP was measured based on fluorescent imaging at indicated time points (hours after V-M8 morpholino or M9 injection) for various genetic backgrounds (GFP, GFP::Rz* or GFP::Rz). All values are displayed as mean ± s.e.m. *n* ≥ 19 animals. The exact number of animals is provided as source data. **i** The percentage of GFP-Rz animals containing normal mitochondrial was quantified at indicated time points after V-M8 morpholino or M9 injection. *n* ≥ 26 animals. The exact number of animals is provided as source data. Arrows in (**b**) and (**e**): enlarged mitochondria. Source data are provided as a Source Data file.

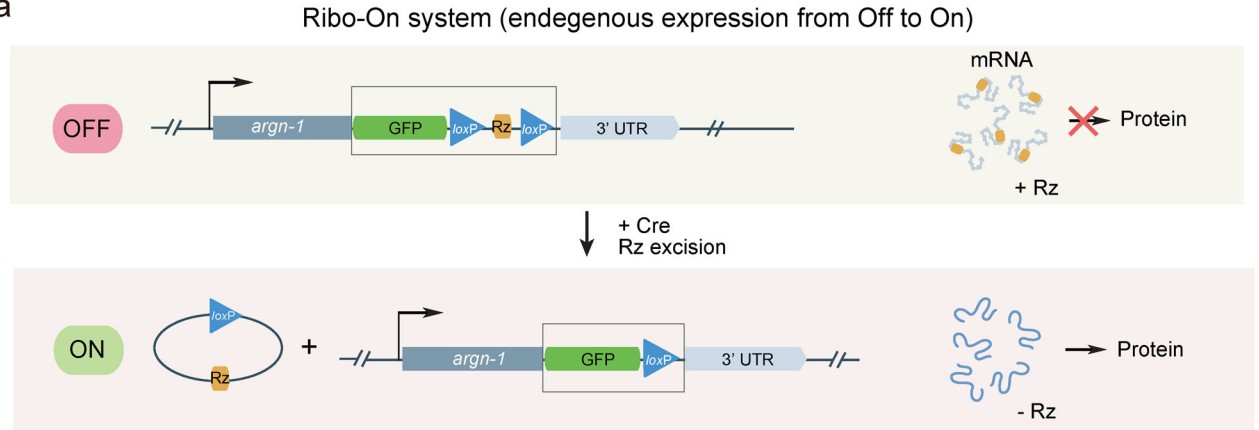

level. However, these methods may induce off-target gene regulation[35–37].

Here, we only performed several proof-of-principle experiments to prove that the Ribo-Off and Ribo-On are valuable tools for manipulating endogenous gene expression. Theoretically, these methods could be further developed for other purposes. First, one can insert tandem ribozymes flanked by loxP sites and

FRT sites to the 3' UTR region of target genes and use different promoters to drive the expression of the Cre and FLP recombinase. Thus, endogenous gene expression will be only induced in the overlapping tissue of two tissue-specific promoters or in the right tissue and at the right time when a tissue-specific promoter and a heat-shock promoter are used in combination[38]. Second, with light-induced Cre expression, a target gene might be

**Fig. 6 The Ribo-On system enables spatiotemporal turn-on of endogenous gene expression. a** Schematic representing how ribozyme/Cre-loxp systems conditionally turn on gene expression. **b** The expression of ARGN-1::GFP in the epidermis and the mitochondrial morphology in one-day-old adult worms under different genetic backgrounds. Cre expression was spatially controlled by tissue-specific promoters or temporally induced by heat-shock promoter, respectively. Scale bars: 5 μm. Arrows: enlarged mitochondria. **c** Quantification of ARGN-1::GFP expression in the epidermis under different conditions, as shown in (**b**). Each dot represents a single worm, $n \geq 25$. The exact number of animals is provided as source data. All data are presented as mean ± s.e.m. $*p < 0.05$. $****p < 0.0001$ (one-way ANOVA with the Tukey correction). **d** Quantifications of the proportion of animals containing normal, intermediate defective or severe defective morphology of mitochondrial in the epidermis under different conditions as shown in (**b**). The number of animals quantified was shown above each column. Source data are provided as a Source Data file.

converted into a light-sensitive allele with a high spatiotemporal resolution[39]. Third, if we use loxP or FRT sites with reduced recombination efficiency, we will be able to generate mosaic animals in which endogenous gene expression or inactivation is achieved in only one or a small number of cells[40].

Our Ribo-Off and Ribo-On systems also have limitations. First, for conditional knockout of endogenous gene expression, Ribo-Off may generate partial loss-of-function phenotypes as the normal mRNAs and proteins can last for several hours or even longer. This is a common issue for other technologies which regulate gene expression at the DNA or RNA level[41–43]. Second, it is easy to control gene expression if one gene generates several isoforms with identical C-terminus. However, if the transcripts are ended with different C-terminus, Ribo-Off and Ribo-On will only regulate one specific isoform, but not others. In this case, floxed TTS cassette can be used if the transcripts share the same start codon. Third, compared to RNAi and CRISPRi, the throughput of Ribo-Off is low. Thus, the former approaches are applicable for doing large-scale screens and Ribo-Off is suitable for studying biological functions of specific candidate genes isolated from the screens. Last, for the conditional Ribo-Off experiments, we used the *let-858* transcriptional termination signal, a commonly used 3' UTR in *C. elegans* expression vectors, which has been validated to be an effective transcriptional terminator[30]. This makes it a popular choice for manipulating gene expression. Theoretically, any 3' UTR that has been validated to effectively terminate transcription can be utilized in the design of a conditional Ribo-off system. Replacing the endogenous 3' UTRs with *let-858* 3' UTR for *argn-1* and *bicd-1* did not affect mitochondrial morphology and PVD dendrite development, respectively, suggesting that *let-858* 3' UTR effectively supports endogenous protein expression (Figs. 3b–d and 4b, c). However, it should be emphasized that for some genes, the regulatory elements within their 3' UTRs may be crucial in controlling their spatiotemporal expression profiles. Thus, in such cases, we propose utilizing the 3' UTRs of the target genes.

In this study, two methods were developed to achieve Ribo-On: combining ribozyme with morpholino or Cre-loxP. However, using morpholino injection only partially restored gene expression (approximately 30 to 60% of the wild-type level) by altering the structure of T3H38 to prevent self-cleavage. Additionally, a high dose of morpholino may cause toxicity in *C. elegans*. In contrast, combining ribozyme with Cre-loxP resulted in an efficient switch from "Off" to "On", where the "Off" state depends on the high self-cleavage efficiency of T3H38. Once Cre was expressed in specific tissue, the target gene's expression was restored to near-normal levels after T3H38 was removed. The efficiency of Cre-loxP has been confirmed in numerous studies, making it a reliable method for achieving the "On" state[44,45]. Therefore, we suggest utilizing ribozyme and Cre-loxP as the primary method for implementing Ribo-On control.

The Ribo-On/Off system relies on gene editing-mediated homology-directed repair (HDR) for ribozyme cassette insertion. In this study, two strategies, namely *unc-119*-based positive selection and *dpy-10*-based co-CRISPR, were designed to achieve

relatively high knock-in efficiency of ribozyme cassettes at the endogenous locus[13,46]. To ensure that both targeted alleles of all animals and cells contain ribozyme cassette insertion, PCR-based genotyping was performed on the progenies derived from self-fertilization. Theoretically, non-specific, off-target insertion of the ribozyme cassette should be rare. Nevertheless, to minimize the possibility of such unwanted insertion, the ribozyme knock-in strain can be outcrossed with the wild-type animals several times before use, and multiple independent insertion strains should be analyzed for phenotypic analysis.

To ensure effective regulation of gene expression, it is crucial to carefully consider the ribozyme insertion site, as ribozyme is an RNA structure that can be influenced by its surrounding sequence. Alternative sites for ribozyme insertion include the 5' and 3' UTRs, as well as exons and introns. Previous studies have confirmed the effectiveness of assembling ribozyme in the C-terminal of the target gene (located in 3' UTR) for controlling the expression of exogenous genes[11]. In this study, all insertion sites were in the 3' UTR. Out of the 15 genes tested, 14 were effectively regulated by the ribozyme, demonstrating the suitability of 3' UTR as a viable site for ribozyme insertion. Previously, the use of ribozyme to artificially manipulate endogenous gene expression was limited, with only one prior study on the parasite *Leishmania*, in which the ribozyme was also inserted into the C-terminal of the target gene[12]. To identify the optimal insertion sites for ribozyme, future studies are required to explore the efficiencies of inserting ribozyme into various locations.

Here we only tested Ribo-Off and Ribo-On for endogenous genes in *C. elegans*. We speculate that similar approaches are likely working in other organisms, including mammals. Thus, our newly developed ribozyme-dependent endogenous gene regulation methods will expand the toolbox for studying gene functions in space and time in vivo.

## Methods

**C. elegans strains and genetics**. *C. elegans* strains were cultured on nematode growth media (NGM) plates seeded with OP50 and maintained at 20 °C[47]. The N2 Bristol strain was used as the wild-type strain. Strains that stably express Cre recombinase driven by various tissue-specific or heat-shock promoters) used in this study were obtained from CGC and Dr. Shohei Mitani's lab[43,45]. *scav-3(tm3659)*[15], *sax-7(nj48)*[16], *drp-1(tm1108)*[19], *eff-1(hy21)*[20], *dma-1(wy686)*[16], *dhgd-1(tm6671)*[18], *hphd-1(ok3580)*[18], *unc-86(e1416)*[26], *mec-3(e1338)*[25], *hpo-30(ok2047)*[27], *kpc-1(gk8)*[28], *rab-10(ok1494)*[29], *mnr-1(wy758)*[16] and *lect-2(ok2617)*[23] are null or strong loss-of-function alleles reported previously. *argn-1(zac427)* is a deletion mutant generated via CRISPR/Cas9 genome editing (a 191 bp genomic DNA fragment is deleted, flanked by 5'-AAAAATCATAGAAGAGGTGA-3' and 5'-GTGGAGTGGAGTAGC-CACCG-3'). It is likely a strong loss-of-function or null allele based on the severely enlarged mitochondria phenotype. *bicd-1(zac51)* was isolated from a forward genetic screen for dendrite abnormal mutants and the details will be published elsewhere. *zjuSi47[Pcol-19>mito::mKate2]*[48], *zjuSi48[Pcol-19>tomm-20::gfp]*[48], *yqIs157[Psemo-1::mito-gfp]*[18], *qxIs257[Pced-1::nuc-1::mCherry]*[15], *wyIs592 [ser2 prom3>myr-gfp]*[23], *wyIs594[ser2prom3>myr-gfp]*[49] and *wyIs587[ser2prom3>myr-mCherry]* were used to label morphologies of mitochondria, lysosomes or PVD dendrites.

To generate the single-copy transgene *zacSi13 [Psemo-1::ctns-1::mCherry; unc-119(+)]*, CRISPR/Cas9-mediated homologous recombination was used[13,50]. The mCherry coding sequence, which contains the *unc-119(+)* expressing cassette in one of its introns, was amplified from pMLS291[51]. Briefly, the repair donor plasmid pFJ147, pDD122 *Peft-3::cas9 + U6prom::ttTi5605 sgRNA* (target sequence: 5' atatcagtctgtttcgtaa 3', kindly provided by Dr. Daniel J. Dickinson) and co-injection

markers *Podr-1::rfp (30 ng/µL)*, *Pmyo-2::mCherry (2 ng/µL)* and *Punc-122::rfp (50 ng/µL)* were injected into *unc-119(ed4)* at the young adult stage. The single-copy transgene was identified based on 100% non-unc and without any co-injection marker expression and validated via PCR-based genotyping and Sanger sequencing.

To make gene edited alleles zac343 (*argn-1::gfp*), zac392 (*argn-1::gfp::Rz*), zac393 (*argn-1::gfp::Rz\**), zac386 (*scav-3::gfp::Rz*), zac387 (*scav-3::gfp::Rz\**), zac389 (*scav-3::gfp*), zac311 (*sax-7::gfp*), zac349 (*sax-7::gfp::Rz*) and zac350 (*sax-7::gfp::Rz\**), an *unc-119(+)* positive selection-based CRISPR/Cas9 editing protocol was used, which was similar as the approach used to make *zacSi13*[13,51].

To make gene edited alleles zac467 (*argn-1::gfp::TTS::loxP::Rz*), zac469 (*scav-3::gfp::loxP::TTS::loxP::Rz*), zac465 (*sax-7::gfp::loxP:: TTS::loxP::Rz*), zac476 (*bicd-1::loxP::TTS:: loxP::Rz*), zac565 (*dhgd-1::loxP::Rz::loxP*), zac562 (*hphd-1::loxP::Rz::loxP*), zac567 (*drp-1::loxP::Rz::loxP*), zac587 (*unc-86::loxP::Rz::loxP*), zac572 (*mec-3::loxP::Rz::loxP*), zac490 (*dma-1::loxP::Rz::loxP*), zac561 (*hpo-30::loxP::Rz::loxP*), zac571 (*kpc-1::loxP::Rz::loxP*), zac586 (*rab-10::loxP::Rz::loxP*), zac573 (*mnr-1::loxP::Rz::loxP*), zac474 (*lect-2::loxP::Rz::loxP*), zac527 (*eff-1b::loxP::Rz::loxP*) and zac475 (*argn-1::gfp:: loxP::Rz::loxP*), a modified co-conversion protocol was used[46]. Briefly, the repair donor plasmid with about 600 bp homology arms, pSX524 (*Peft-3::Cas9+ U6prom:: dpy-10 sgRNA*, target sequence: 5' CTACCATAGGCACCACGAG 3', kindly provided by Dr. Suhong Xu) and 2–3 independent *U6prom::sgRNA* plasmids for gene of interest were injected into *N2* at the young adult stage. After 4 days, dumpy or roller F1 animals were picked and grew on OP50-seeded plates (3 F1 animals per plate). PCR-based genotyping was performed to identify successful knock-in editing, and Sanger sequencing was performed to make sure there was no extra mutation. Information on worm strains, plasmids and primers (for making sgRNAs and performing qRT-PCR only) used in this study can be found in Supplementary Tables 1, 2 and 3, respectively. Additional information for primers used for genotyping and plasmid construction will be available upon request.

**Plasmid construction.** To build T3H38 ribozyme constructs, the sequences of active T3H38 (Rz) and inactive T3H38 (Rz\*) were commercially synthesized by Qinlan Biotech[10]. The DNA template donors for HDR-mediated knock-in were generated by assembling homology arms (5' and 3' HA) of around 600 bp and inserts (*gfp*, *gfp::Rz*, *gfp::Rz\**, or *gfp::loxP::Rz::loxP* or *gfp::loxP::let858 TTS::loxP::Rz* using a multi-fragment cloning kit[30,31]. The sequences coding for *loxP::Rz::loxP* or *loxP::let858 TTS::loxP::Rz* were listed in Supplementary Tables 4 and 5. A ~3.2 kb sgRNA expression vector was obtained by cloning the U6 promoter, specific spacer sequence and optimized sgRNA scaffold (E+F) into an ampicillin-resistant backbone vector[52]. A PCR-based Quik-Change cloning method was used to generate plasmids to express sgRNA for gene of interest. Briefly, a reverse primer with 5' phosphate added and a forward primer with target sequence and part of the sgRNA scaffold was used to amplify the entire sgRNA template plasmid. The concentration of the template DNA used in the PCR reaction was 0.25 ng/µL. Two microliters of PCR product was used for ligation by the T4 DNA ligase. The ligation product was transformed into competent cells, and correct sgRNA plasmids were identified via Sanger sequencing. To construct dsRNA-expressing plasmids, L4440 was used as the vector backbone. *sax-7* and *argn-1* cDNAs were amplified from a cDNA library of mixed-stage wild-type *C. elegans* by PCR and then inserted into the vector backbone.

**Quantitative reverse transcription PCR analysis.** Six to eight 6-cm plates of synchronized worms were collected into a 1.5 mL centrifuge tube, supplemented with 200 µL of TRizol (Ambion, 15596018), put in liquid nitrogen, thawed and transferred into liquid nitrogen again, and finally stored in a −80 °C refrigerator until the next step. RNA was extracted using a cell/tissue total RNA extraction kit (YEASEN, 19221ES50) and reverse transcribed using a reverse transcription kit (HiScript III 1st Strand cDNA Synthesis Kit, Vazyme). Real-time qPCR was then performed on a CFX-96 thermal cycler (Bio-Rad) using 2xUniversal SYBR Green Fast qPCR Mix (ABclonal) master mix. The qPCR program was set as follows: 95 °C for 10 min and 40 cycles at 95 °C for 15 s/60 °C for 30 s. All reactions were done at least in triplicate. Data were analyzed using the comparative $2^{-\Delta\Delta Ct}$ method. *ama-1* was used as the reference gene. qRT-PCR primers are listed in Supplementary Table 3.

**RNA interference assay.** *C. elegans* were placed on NGM plates seeded with L4440 (empty vector as a negative control) or bacteria strains expressing gene-specific dsRNAs and cultured at 20 °C. In Fig. 2, L4-satge *zjuSi47[Pcol-19>mito::mKate2]*, *zjuSi48[Pcol-19>tomm-20::gfp]*, *qxIs257[Pced-1::nuc-1::mCherry]* and *wyIs592[ser2prom3>myr-gfp]* transgenic animals were transferred onto plates seeded with L4440 or bacterial strains producing gene-specific double-strand RNAs (from the Vidal RNAi library or constructed by ourself)[53]. F1 animals at the 1-day-old adult (24 h post L4) stage were imaged and analyzed for mito-chondrial morphology/lysosomal morphology/PVD dendrite morphology.

**v-M8 injection.** The vivo-morpholino (v-M8, GENE TOOLS) was dissolved in DMSO (500 µM stock solution). V-M8 was diluted with M9 into a series of solutions of different concentrations. 1-day-old adult animals for *zac343(argn-1::gfp)*, zac392 (*argn-1::gfp::Rz*), zac393 (*argn-1::gfp::Rz\**), zac386 (*scav-3::gfp::Rz*), zac387 (*scav-3::gfp::Rz\**) were injected with v-M8 and imaged 48 h after injection. 1-day-old animals were injected with v-M8 to assess the expression of the endogenous ARGN-1::GFP and SCAV-3::GFP and the corresponding phenotypic analysis. Animals at the L4 stage for zac349 (*sax-7::gfp::Rz*) and zac350 (*sax-7::gfp::Rz\**) were injected with v-M8 and imaged 48 h after injection to measure the expression of the endogenous SAX-7::GFP and quantify the number of PVD dendrite branches.

**Heat-shock induction of Cre expression.** Before heat-shock treatment, the strains with the heat-shock::Cre transgene were carefully maintained in a 15 °C incubator to avoid any Cre expression. To induce Cre expression, L4-stage animals were incubated at a 33 °C water bath for 1 h. Then the animals were put into a 20 °C incubator to recover for 24 h or 48 h or 72 h before confocal imaging.

**Confocal imaging.** Before imaging, worms were immobilized with 20 mM leva-misole on a 4% agar pad. All confocal fluorescent images in this study were acquired on an Olympus IX83 fluorescence microscope equipped with spinning-disk confocal scanner (Yokogawa CSU-W1), U Plan Super Apochromat objectives (10x/0.4, 20x/0.7 and 40x/0.95), a 60 x NA 1.49 oil Apochromat objective, an sCMOS camera (Prime 95B), 488/561 nm lasers (OBIS) and a PIEZO stage (ASI). For PVD dendrites, a region anterior to the PVD cell body was imaged and shown. For organelles (mitochondria and lysosomes), the anterior part of the worm body was imaged and shown. All images were acquired using a 60x objective. ARGN-1::GFP, SCAV-3::GFP, NUC-1::mCherry, CTNS-1::mCherry, MITO::mKate and TOMM-20::GFP were imaged and shown as single-focal-plane images[15,48]. PVD dendrite morphology and SAX-7::GFP were imaged in Z-stack with 1 µm step size, and the maximum intensity projections were shown.

**Quantification for the morphologies of mitochondria, lysosomes and PVD dendrites.** For Figs. 1d, 3d and 5g, Supplementary Figs. S3c, f and S4c, a 220 × 220 µm region was imaged for each animal at the 2-day-old or 3-day-old adult stage. For Figs. 2b, d and 6d, Supplementary Fig. S2b, d, 1-day-old animals were imaged. For mitochondria, "Normal" means either most of the mitochondria were tubules, or tubules and globular structures (with diameters less than 1.5 µm); "Intermediate" means that some mitochondria were enlarged (with diameters more than 1.5 µm but less than 3 µm); "Severe" means that some mitochondria were severely enlarged (with diameters more than 3 µm). For lysosomes, "Normal" means a 220 × 220 µm imaged area contained less than five enlarged lysosomes (defined as lysosomes with a diameter more than 2.7 µm; "Intermediate" means that a unit area contained 5 to 10 enlarged lysosomes; "Severe" means that a unit area contained more than 10 enlarged lysosomes. For PVD development, "Normal" means that a unit area was fully occupied with tertiary and quaternary branches; "Intermediate" means that a unit area was partially covered by those branches; "Severe" means that no tertiary or quaternary branches were formed in a unit area.

**Statistics and reproducibility.** All fluorescent imaging data were analyzed by ImageJ to calculate the mean fluorescence intensity of mCherry and GFP. To measure endogenous protein expression via quantitative fluorescence imaging, five representative regions for each animal were selected to measure the average mean fluorescence intensity, and then the mean intensity of the background signal was subtracted from the mean of the five regions. To quantify the dendrite development phenotypes, numbers of 2°, 3° and 4° dendritic branches were quantified manually in a 100 µm region anterior to the PVD cell body. The parameters were then analyzed in Excel and plotted in GraphPad Prism. GraphPad Prism was used to calculate the mean values, the standard error of the mean (SEM) and the statistical significance using one-way ANOVA with the Tukey correction (statistical significance is indicated as ns, not significant if $p > 0.05$; $^*p < 0.05$; $^{**}p < 0.01$; $^{***}p < 0.001$; $^{****}p < 0.0001$) for most results, except for the comparisons between Ribo-Off and RNAi for *argn-1*, *drp-1* and *scav-3* genes showed in Fig. 2b. In these cases, χ2 test was used and statistical significance was indicated as $^*p < 0.05$; $^{**}p < 0.01$.

Sample sizes were determined based on previous publications in which similar experiments were performed, and the number of animals used in each experiment can be found in the figures/figure captions/Source Data file (Supplementary Data 1). For most of the experiments, measurements were taken from distinct samples except for the qRT-PCR experiments (each sample was tested three times and three biological repeats were measured as shown in Fig. 1f). All experiments were performed at least two times with similar results.

**Reporting summary.** Further information on research design is available in the Nature Portfolio Reporting Summary linked to this article.

## Data availability
The data that support this study are available from the corresponding authors upon reasonable request. Key plasmids will be deposited to Addgene. Source data are provided with this paper as Supplementary Data 1.

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

## Acknowledgements

We thank Drs. Kang Shen (Stanford University), Xiaochen Wang (Chinese Academy of Sciences), Chonglin Yang (Yunnan University), Suhong Xu (Zhejiang University), Zhiping Wang (Zhejiang University), Daniel J. Dickinson (UT Austin), Martin Harterink (Utrecht University) and Erik M. Jorgensen (University of Utah) for kindly sharing plasmids and *C. elegans* strains, and Dr. Lijun Kang (Zhejiang University) for sharing some equipment. This work was supported by the National Natural Science Foundation of China grants (3217144 to B.C. and 31970919 to W.Z.) and the National Key R&D Program of China (2021YFC2700904) to B.C. Some strains were provided by the CGC, which is funded by the NIH Office of Research Infrastructure Programs (P40 OD010440), and the MITANI Lab through the National Bio-Resource Project of the MEXT, Japan.

## Author contributions

J.F., J.W., Y.W., and X.L. performed experiments and analyzed data. W.Z. and B.C. conceived and supervised the project and wrote the manuscript with input from all authors.

## Competing interests

The authors declare no competing interests.
