## [Peer Review File · Communications Biology]

Reviewers' comments:

Reviewer #1 (Remarks to the Author):

This is a great manuscript describing a useful technology to be added into the *C. elegans* toolbox, and potentially can be extended to other organisms in the future. I recommend publication with the following minor suggestions/questions:

1) In the Ribo-off system, Fig 1a, since 3'UTR contains a lot of regulatory elements, would the substitution of endogenous 3'UTR with let-858 terminator change the function or regulation of the target gene? It was shown by the authors that this does not affect the gene tested. How about other genes that will be used in the future? Why not duplicate the endogenous 3'UTR within the loxP? Could the authors comment on this?

2) A typo: line 234, "CRISPR" misspelled as "CRIPSR".

Reviewer #2 (Remarks to the Author):

In this manuscript, Fang et al. describe a very useful system to control the expression of endogenous genes via manipulation of the activity of a ribozyme. The approach makes use of the T3H38 ribozyme, a previously reported self-cleaving ribozyme that displays very high activity. Fang and colleagues insert the ribozyme into the 3' UTR of mRNAs in *C. elegans* to trigger rapid mRNA degradation following cleavage. They use Cre recombinase and morpholinos to modulate the activity of the ribozyme. The former strategy allows the gene expression to be turned off (Ribo-Off) or turned on (Ribo-On) in response to heat or tissue-specific promoters. The latter strategy allows gene expression to be turned on in a dose-dependent manner by morpholinos that hybridize with the ribozyme. The authors provide a detailed set of experiments documenting the performance of the system, the results appear robust, and the system can be applied to multiple endogenous mRNAs. While ribozymes have been used for modulating gene expression in multiple other papers, the approach should be broadly useful for studies of *C. elegans* and is likely to be extended to other organisms. I recommend publication of this work once the points below have been addressed:

1. In the morpholino experiments, there appears to a substantial reduction in the GFP fluorescence from the regulated gene for the active ribozyme at 400 μ M morpholino (Fig. 4e). What is the cause of this effect? Does 400 μ M morpholino cause a similar effect for the deactivated ribozyme?

2. Error bars and individual points: The figures in the paper use the standard error in the mean (s.e.m.) for the error bars. Based on the spread of experimental measurements, the standard deviation would seem to be a more reasonable representation of the spread in the data. Also, the error bars in Figure 1f *may* be using the s.d. rather than s.e.m. based on the error bar range and the spread of individual points -- the authors should check what statistical formula they are using in 1f. Individual data points are also not provided for Figs. 3c, 4h-i, S1, S3, S4, and S5. These should be included in the figures, particularly since the error bars do not reflect the true spread of the data when s.e.m. is used.

3. The studies with Cre describe the insertion to be "gfp::loxP::let-858 transcriptional termination sequence (stop)::loxP::T3H38". Is there a stop codon included in this insert? The use of the term "stop" is confusing if it is only meant to specify a transcriptional terminator.

4. On line 79, the authors describe the mRNA with the deactivated ribozyme as a "stabilized mRNA". Does the deactivated ribozyme actually stabilize the mRNA compared to the endogenous (unmodified) version? If not, I suggest changing this wording.

5. Line 130: "Specific expression" should be "Specific expression".

Reviewer #3 (Remarks to the Author):

In this manuscript, the authors developed a novel method to manipulate the "on" and "off" state of endogenous genes in space and time in *C. elegans*, by inserting a ribozyme sequence into the 3'UTR with CRISPR/Cas9. The self-cleavage of ribozyme achieves gene knockout, making an "off" switch. Adding antisense oligo can interrupt ribozyme function and rescue gene repression to a certain extent, making an "on" switch. The method can be combined with the Cre-loxP system to achieve spatiotemporal regulation of endogenous gene expression. The method is of potential interest for *C. elegans* research. However, more comparison with the state-of-the-art methods is needed to increase the impact of this method.

Major questions:

1. One major question related to the method is generality and whether it can be applied to many genes. Considering ribozyme is an RNA structure that can be affected by the sequence context that it is inserted, the method may not work for other genes. Secondary structure in the target RNA may disrupt the ribozyme function and make the method less useful. In this case, the authors should evaluate how other genes may or may not use their approach. Furthermore, insertion sites can be critical for the performance. How do other researchers choose the insertion sites to ensure the best switch behavior?
2. The manuscript provided evidence of gene repression and activation to affect important phenotypes in *C. elegans* using their new method. However, it is unclear how the efficiency compares with previous methods of endogenous gene regulation, including RNA interference and CRISPR interference. Does the gene knockdown reach the same level as that can be achieved by the state-of-the-art methods in *C. elegans*? A side-by-side comparison is needed.
3. The Ribo-on by adding antisense oligo to reactivate the gene cannot bring the expression back to wildtype level. For certain genes, there is a remarkable reduction in gene expression level (60-80%) in the "on" state. Although the authors proved that such mild activation is sufficient to reverse some phenotypes back to wildtype level, it remains unclear whether the method can be applied to many *C. elegans* genes, especially those that are sensitive to change in doses. Are there selection criteria for genes that are suitable for regulation using this approach?
4. CRISPR/Cas9 cutting in the new method can lead to non-homologous end joining (NHEJ) and homologous directed repair (HDR). What is the ratio of animals or cells that have both alleles successfully inserted with ribozyme? What is the ratio of NHEJ and HDR? Does the possibility of NHEJ affect the conclusion in the manuscript? Some discussion is needed.
5. In Figure 4, one injection of morpholino can sustain gene activation for around 72h, which is able to rescue phenotypes like mitochondria morphology. However, for biological phenotypes that requires a longer-period of gene expression, can serial dosing of morpholino maintain gene activation for more sustained period, without reaching the point where gene activation is reversed by high concentration of morpholino (400uM)?
6. In the introduction, the authors discussed potential off-target effects of previous methods like CRISPR interference. However, the new method also relies on CRISPR/Cas9 for the knockin of ribozyme. How is the issue of off-target effect addressed with the new method? Are there potential off-target cutting or insertion of ribozyme? Rewording the introduction is needed since the specificity may not be their advantage.
7. Since the authors did not test the performance of this method in other living systems, the claim in the title should also be narrowed down accordingly to "in *C. elegans*".

Other issues:

1. The first row of Figure1C is poorly labeled. Should indicate what the mutant strains *zac427*, *tm3659*, *nj48* are.

2. In Figure 1E, the level of Sax-7 is significantly changed with the addition of inactivated ribozyme, so the text "Sax-7 protein remained almost unchanged" is not very accurate.
3. Since the insertion of inactivated ribozyme already leads to reduction in mRNA and protein level of the endogenous genes, the authors should discuss potential reasons. Is that due to a faster degradation rate, or is there some residual activity in the ribozyme? Or other reasons?
4. In Figure 4, the addition of 400uM Morpholino reversed the gene activation effect. The authors should discuss potential reason of this repression.

Response to the referees' comments

Reviewer #1 (Remarks to the Author):

This is a great manuscript describing a useful technology to be added into the *C. elegans* toolbox, and potentially can be extended to other organisms in the future. I recommend publication with the following minor suggestions/questions:

Response: We are grateful to the reviewer for the positive evaluation of our work. We greatly appreciate the following comments and have made corresponding modifications according to the reviewer's suggestions.

1) In the Ribo-off system, Fig 1a, since 3'UTR contains a lot of regulatory elements, would the substitution of endogenous 3'UTR with *let-858* terminator change the function or regulation of the target gene? It was shown by the authors that this does not affect the gene tested. How about other genes that will be used in the future? Why not duplicate the endogenous 3'UTR within the loxP? Could the authors comment on this?

Response: We appreciate the reviewer's advice. According to the reviewer's suggestion, we have added the following comments in the discussion section:

"Last, for the conditional Ribo-Off experiments, we used the *let-858* transcriptional termination signal, a commonly used 3' UTR in *C. elegans* expression vectors, which has been validated to be an effective transcriptional terminator. This makes it a popular choice for manipulating gene expression. Theoretically, any 3' UTR that has been validated to effectively terminate transcription can be utilized in the design of a conditional Ribo-off system. Replacing the endogenous 3' UTRs with *let-858* 3' UTR for *argn-1* and *bicd-1* did not affect mitochondrial morphology and PVD dendrite development, respectively, suggesting that *let-858* 3' UTR effectively supports endogenous protein expression (Figs. 3b-3d and 4b-4c). However, it should be emphasized that for some genes, the regulatory elements within their 3' UTRs may be crucial in controlling their spatiotemporal expression profiles. Thus, in such cases, we propose utilizing the 3' UTRs of the target genes."

2) A typo: line 234, "CRISPR" misspelled as "CRIPSR".

Response: Corrected. Thanks!

Reviewer #2 (Remarks to the Author):

In this manuscript, Fang et al. describe a very useful system to control the expression of endogenous genes via manipulation of the activity of a ribozyme. The approach makes use of the T3H38 ribozyme, a previously reported self-cleaving ribozyme that displays very high activity. Fang and colleagues insert the ribozyme into the 3' UTR of mRNAs in *C. elegans* to trigger rapid mRNA degradation following cleavage. They use Cre recombinase and morpholinos to modulate the activity of the ribozyme. The former strategy allows the gene expression to be turned off (Ribo-Off) or turned on (Ribo-On) in response to heat or tissue-specific promoters. The latter strategy allows gene expression to be turned on in a dose-dependent manner by morpholinos that hybridize with the ribozyme. The authors provide a detailed set of experiments documenting the performance of the system, the results appear robust, and the system can be applied to multiple endogenous mRNAs. While ribozymes have been used for modulating gene expression in multiple other papers, the approach should be broadly useful for studies of *C. elegans* and is likely to be extended to other organisms. I recommend publication of this work once the points below have been addressed:

Response: We thank the reviewer for recognizing the significance of the Ribo-On/Off technology and providing clear guidance on how to improve our manuscript. In response to the reviewer's suggestions, we conducted additional experiments and made corresponding revisions to further improve the quality of our manuscript. Our detailed point-by-point responses to the reviewer's comments are presented below.

1. In the morpholino experiments, there appears to a substantial reduction in the GFP fluorescence from the regulated gene for the active ribozyme at 400 μ M morpholino (Fig. 4e). What is the cause of this effect? Does 400 μ M morpholino cause a similar effect for the deactivated ribozyme?

Response: We thank the reviewer for pointing this out. To address the reviewer's question, we injected 400 μ M

morpholino into both *argn-1::gfp* and *argn-1::gfp::inactive ribozyme* strains. We found that the expression levels of *argn-1::gfp* and *argn-1::gfp::inactive ribozyme* were not reduced (Data 1), suggesting that a high concentration of morpholinos likely does not generally inhibit gene transcription. The cause of this effect is not clear currently.

Data 1. Effect of injecting 400 μ M morpholino on the expression of ARGN-1::GFP and ARGN-1::GFP::inactive ribozyme

2. Error bars and individual points: The figures in the paper use the standard error in the mean (s.e.m.) for the error bars. Based on the spread of experimental measurements, the standard deviation would seem to be a more reasonable representation of the spread in the data. Also, the error bars in Figure 1f *may* be using the s.d. rather

than s.e.m. based on the error bar range and the spread of individual points -- the authors should check what statistical formula they are using in 1f. Individual data points are also not provided for Figs. 3c, 4h-i, S1, S3, S4, and S5. These should be included in the figures, particularly since the error bars do not reflect the true spread of the data when s.e.m. is used.

Response: We appreciate the reviewer's suggestions and apologize for any confusion caused by the error bars in Fig. 1f, which did indeed represent standard deviation (s.d.). We have now showed s.e.m. for Fig. 1f to make it consistent with other results. Following the reviewer's suggestions, we have added individual data points in Figs. 2d, 4c, S2e, S5b and S6b to show the spread of the data. We have not displayed data points in Figs. 5h and S4b to preserve their visual appeal. In Figs. 4i and S3c, each data point represents a single value indicating the percentage of animals with normal organelle morphology, with sample sizes greater than 30 animals.

3. The studies with Cre describe the insertion to be "gfp::loxP::let-858 transcriptional termination sequence (stop)::loxP::T3H38". Is there a stop codon included in this insert? The use of the term "stop" is confusing if it is only meant to specify a transcriptional terminator.

Response: We thank the reviewer for pointing out this issue. We inserted the 3' UTR of *let-858* to terminate transcription, but not a stop codon to terminate translation. Following this reviewer's suggestion, we now use "TTS" rather than "stop" to stand for the *let-858* transcriptional termination sequence.

4. On line 79, the authors describe the mRNA with the deactivated ribozyme as a "stabilized mRNA". Does the deactivated ribozyme actually stabilize the mRNA compared to the endogenous (unmodified) version? If not, I suggest changing this wording.

Response: We are grateful to the reviewer for bringing this issue to our attention. As suggested by the reviewer, the term 'stabilized mRNA' is not appropriate to describe the situation where mRNA remains uncut due to ribozyme inactivation. Therefore, we have replaced it with 'uncleaved mRNA'. We hope that the reviewer finds this modification to be suitable.

5. Line 130: "Specifical expression" should be "Specific expression".

Response: Corrected. Thanks!

Reviewer #3 (Remarks to the Author):

In this manuscript, the authors developed a novel method to manipulate the “on” and “off” state of endogenous genes in space and time in *C. elegans*, by inserting a ribozyme sequence into the 3'UTR with CRISPR/Cas9. The self-cleavage of ribozyme achieves gene knockout, making an “off” switch. Adding antisense oligo can interrupt ribozyme function and rescue gene repression to a certain extent, making an “on” switch. The method can be combined with the Cre-loxP system to achieve spatiotemporal regulation of endogenous gene expression. The method is of potential interest for *C. elegans* research. However, more comparison with the state-of-the-art methods is needed to increase the impact of this method.

Response: We thank the reviewer for spending time reviewing our work and the positive assessment of Ribo-On/Off systems. Following the reviewer's constructive suggestions, we performed new experiments and additional data analysis, added new figures (Figs. 2, S1 and S2), and modified some contents and claims accordingly in the revised manuscript as suggested by the reviewer. We hope that the reviewer could find this revised manuscript more solid. Our point-by-point responses to the reviewer's comments are stated below.

Major questions:

1. One major question related to the method is generality and whether it can be applied to many genes. Considering ribozyme is an RNA structure that can be affected by the sequence context that it is inserted, the method may not work for other genes. Secondary structure in the target RNA may disrupt the ribozyme function and make the method less useful. In this case, the authors should evaluate how other genes may or may not use their approach. Furthermore, insertion sites can be critical for the performance. How do other researchers choose the insertion sites to ensure the best switch behavior?

Response: We appreciate the reviewer for raising this critical issue. Although ribozyme has been utilized to control the expression of exogenous genes, its use for regulating endogenous gene expression has remained

limited. In this study, we have successfully inserted the T3H38 ribozyme sequence between the stop codon and the 3' UTRs of five endogenous genes (*argn-1*, *scav-3*, *sax-7*, *dma-1* and *lect-2*) and subsequently tested ten additional genes (*dhgd-1*, *hphd-1*, *drp-1*, *unc-86*, *mec-3*, *hpo-30*, *mnr-1*, *rab-10*, *kpc-1* and *eff-1*) during revision. As shown in Fig. S2, our updated results demonstrated that T3H38 ribozyme efficiently downregulated gene expression in 14 out of the 15 genes we tested, with the exceptions of the *kpc-1* gene (possibly due to the sequence context that the ribozyme is inserted as this reviewer pointed out). For those 14 genes we tested, inserted the ribozyme sequence between the stop codon and the 3' UTR generated null-like loss-of-function alleles, suggesting that the currently used insertion strategy works with a relatively high efficiency. Please see line 104-line 122 for the description of the new results. We agree with this reviewer that the insertion site of ribozyme is a critical factor to consider. However, we were unable to examine more alternative insertion sites in this study due to time constraints.

We have also added a new paragraph in the Discussion to address this issue:

“To ensure effective regulation of gene expression, it is crucial to carefully consider the ribozyme insertion site, as ribozyme is an RNA structure that can be influenced by its surrounding sequence. Alternative sites for ribozyme insertion include the 5' and 3' UTRs, as well as exons and introns. Previous studies have confirmed the effectiveness of assembling ribozyme in the C-terminal of the target gene (located in 3' UTR) for controlling the expression of exogenous genes. In this study, all insertion sites were in the 3' UTR. Out of the 15 genes tested, 14 were effectively regulated by the ribozyme, demonstrating the suitability of 3' UTR as a viable site for ribozyme insertion. Previously, the use of ribozyme to artificially manipulate endogenous gene expression was limited, with only one prior study on the parasite *Leishmania*, in which the ribozyme was also inserted into the C-terminal of the target gene. To identify the optimal insertion sites for ribozyme, future studies are required to explore the efficiencies of inserting ribozyme into various locations.”

Fig. S2 Additional endogenous genes inactivated by ribozyme knockin.

2. The manuscript provided evidence of gene repression and activation to affect important phenotypes in *C. elegans* using their new method. However, it is unclear how the efficiency compares with previous methods of endogenous gene regulation, including RNA interference and CRISPR interference. Does the gene knockdown reach the same level as that can be achieved by the state-of-the-art methods in *C. elegans*? A side-by-side comparison is needed.

Response: We appreciate the valuable comments from this reviewer. Indeed, comparing ribozyme-mediated interference with existing methods, including RNA interference (RNAi) and CRISPR interference (CRISPRi), is essential for evaluating its potential utility. While CRISPRi has limited usage in *C. elegans* (probably due to its low efficiency), RNAi is the most widely used method for gene knockdown in this model organism. Therefore, we performed a side-by-side comparison of ribozyme-mediated interference with RNAi for six genes. The new results have been presented in Fig. 2, and we have added a paragraph to describe these findings into the result part as follows:

“We next compared the knockdown efficiency of Ribo-Off and RNA interference for six genes based on the severity of their loss-of-function phenotypes. Our results showed that inserting the ribozyme into *argn-1*, *drp-1* and *scav-3* led to almost 100% of animals displaying severely enlarged mitochondria or lysosome phenotypes, whereas gene-specific dsRNAs resulted in intermediate phenotypes in small but significant proportions of animals (Fig. 2a-2b). Loss of endogenous *eff-1* caused ectopic branching in the lateral regions, while loss of *dma-1* or *sax-7* resulted in reduced higher-order branch formation (tertiary and quaternary branches). The whole-body Ribo-Off alleles of *eff-1*, *dma-1* and *sax-7* phenocopied the previously reported strong loss of function or null alleles, while *eff-1* (RNAi), *dma-1*(RNAi) and *sax-7* (RNAi) animals showed less severe dendrite branching defects (Fig. 2c-2d). Overall, our findings suggest that Ribo-Off outperformed RNAi, at least for the six genes we tested.”

Fig. 2 A comparison of ribozyme-mediated gene inactivation with RNA interference.

3. The Ribo-on by adding antisense oligo to reactivate the gene cannot bring the expression back to wildtype level. For certain genes, there is a remarkable reduction in gene expression level (60-80%) in the “on” state. Although the authors proved that such mild activation is sufficient to reverse some phenotypes back to wildtype level, it remains unclear whether the method can be applied to many *C. elegans* genes, especially those that are sensitive to change in doses. Are there selection criteria for genes that are suitable for regulation using this approach?

Response: We thank the reviewer for bring up this concern, and we have addressed it in the Discussion section as follows:

“In this study, two methods were developed to achieve Ribo-On: combining ribozyme with morpholino or Cre-loxP. However, using morpholino injection only partially restored gene expression (approximately 30% to 60% of the wild-type level) by altering the structure of T3H38 to prevent self-cleavage. Additionally, a high dose of morpholino may cause toxicity in *C. elegans*. In contrast, combining ribozyme with Cre-loxP resulted in an efficient switch from “Off” to “On”, where the “Off” state depends on the high self-cleavage efficiency of T3H38. Once Cre was expressed in specific tissue, the target gene’s expression was restored to near-normal levels after T3H38 was removed. The efficiency of Cre-loxP has been confirmed in numerous studies, making it a reliable method for achieving the “On” state. Therefore, we suggest utilizing ribozyme and Cre-loxP as the primary method for implementing Ribo-On control.”

4. CRISPR/Cas9 cutting in the new method can lead to non-homologous end joining (NHEJ) and homologous directed repair (HDR). What is the ratio of animals or cells that have both alleles successfully inserted with ribozyme? What is the ratio of NHEJ and HDR? Does the possibility of NHEJ affect the conclusion in the manuscript? Some discussion is needed.

Response: The reviewer raised an important question regarding the generation of Ribo-ON/Off strains using CRISPR/Cas9 editing. To explain our strategies more clearly, we have created a schematic diagram (Fig. S1) outlining the experimental principles we used to obtain ribozyme knockin worms.

1) For ribozyme cassettes that do not contain a fluorescent protein, such as 5’ HA-ribozyme-3’ HA, we injected a plasmid mixture containing a Cas9 expressing plasmid, 1-3 plasmids to express site-specific sgRNAs, a plasmid as the HDR donor, and a plasmid to express a sgRNA targeting *dyp-10* gene, into wild-type worms. By

screening for dumpy or roller animals (caused by *dpy-10* loss-of-function and gain-of-function, respectively), we could identify worms with precise insertion more efficiently. Animals with the correct insertion of the ribozyme was isolated by PCR-based genotyping and further verified through Sanger sequencing.

2) For ribozyme cassettes that contain a fluorescent protein, such as 5' HA-GFP-ribozyme-3' HA, we inserted *unc-119* expression cassette into the intron region of *gfp*. The plasmid mixture, including a Cas9 expressing plasmid, 1-3 sgRNA expressing plasmids and a HDR donor, was then injected into the *unc-119* mutants. Successful injection was indicated by progenies showed normal movement behavior. We then selectively picked out the non-Unc F2 generation worms and confirmed the correct *gfp*-ribozyme insertion through PCR-based genotyping and Sanger sequencing.

The above two methods directly identified the animals in which HDR occurred successfully and precisely. Animals with mutations induced by NHEJ were not isolated in the genotyping step and were discarded. We have added a paragraph to discuss the strategies for ribozyme knockin as follows:

“The Ribo-On/Off system relies on gene editing-mediated homology-directed repair (HDR) for ribozyme cassette insertion. In this study, two strategies, namely *unc-119*-based positive selection and *dpy-10*-based co-CRISPR, were designed to achieve relatively high knockin efficiency of ribozyme cassettes at the endogenous locus. To ensure that both targeted alleles of all animals and cells contain ribozyme cassette insertion, PCR-based genotyping was performed on the progenies derived from self-fertilization. Theoretically, non-specific, off-target insertion of the ribozyme cassette should be rare. Nevertheless, to minimize the possibility of such unwanted insertion, the ribozyme knockin strain can be outcrossed with the wild-type animals for several times before use, and multiple independent insertion strains should be analyzed for phenotypic analysis.”

Fig S1. Strategies to generate ribozyme-knockin animals via CRISPR-mediated homology-directed repair.

5. In Figure 4, one injection of morpholino can sustain gene activation for around 72h, which is able to rescue phenotypes like mitochondria morphology. However, for biological phenotypes that requires a longer-period of gene expression, can serial dosing of morpholino maintain gene activation for more sustained period, without reaching the point where gene activation is reversed by high concentration of morpholino (400uM)?

Response: We thank this reviewer for pointing this out. Zhang et al. have successfully utilized T3H38 to regulate exogenous luciferase expression in mice. By performing multiple injections of morpholino, luciferase expression could be repeatedly induced over a period of at least 43 weeks (PMID: 31873216). Therefore, we think that the approach of serial morpholino dosing can be employed to sustain gene activation for an extended duration in *C. elegans*, too. We agree that serial dosing of morpholino possibly causes morpholino accumulation, and if it reaches a high concentration, it may fail to induce gene activation. Thus, when we want to achieve a longer-period of gene expression, we recommend using the Cre-loxP-mediated Ribo-On system.

6. In the introduction, the authors discussed potential off-target effects of previous methods like CRISPR interference. However, the new method also relies on CRISPR/Cas9 for the knockin of ribozyme. How is the issue of off-target effect addressed with the new method? Are there potential off-target cutting or insertion of ribozyme? Rewording the introduction is needed since the specificity may not be their advantage.

Response: We appreciate the reviewer for bringing this to our attention. In our study, we utilized CRISPR/Cas9 technology to achieve precise homologous recombination repair and obtained worms with precise insertion of the ribozyme cassette. Validation of the precise insertion was done through PCR and DNA sequencing. To minimize background mutations caused by CRISPR/Cas9, *C. elegans* strains carrying the precisely inserted ribozyme cassette can be outcrossed for multiple rounds before use and multiple independent knockin strains can be analyzed. Thus, our approach ensures the generation of edited worms with high accuracy for Ribo-On/Off. In contrast, CRISPR interference relies on the base pairing between the sgRNA and the target sequence. During this process, mismatch at the 5' end can be tolerated (PMID: 22745249), which may cause off-target knockdown.

We have explained this advantage of Ribo-On/Off in the Discussion as follows:

“The Ribo-On/Off system relies on gene editing-mediated homology-directed repair (HDR) for ribozyme cassette insertion. In this study, two strategies, namely *unc-119*-based positive selection and *dpy-10*-based co-CRISPR, were designed to achieve relatively high knockin efficiency of ribozyme cassettes at the endogenous locus. To ensure that both targeted alleles of all animals and cells contain ribozyme cassette insertion, PCR-based genotyping was performed on the progenies derived from self-fertilization. Theoretically, non-specific, off-target insertion of the ribozyme cassette should be rare. Nevertheless, to minimize the possibility of such unwanted insertion, the ribozyme knockin strain can be outcrossed with the wild-type animals for several times before use, and multiple independent insertion strains should be analyzed for phenotypic analysis.”

7. Since the authors did not test the performance of this method in other living systems, the claim in the title should also be narrowed down accordingly to “in *C. elegans*”.

Response: We thank the reviewer for this suggestion and have changed the original title to “Ribo-On and Ribo-Off: efficient manipulation of endogenous gene expression using a self-cleaving ribozyme in *C. elegans*”.

Other issues:

1. The first row of Figure1C is poorly labeled. Should indicate what the mutant strains *zac427*, *tm3659*, *nj48* are.

Response: We thank the reviewer for pointing this out. *zac427*, *tm3659*, *nj48* are whole-body knockout alleles. They are strong loss of function or null alleles and used for comparisons between conventional KO and Ribo-Off. We have added the description of allele information of all the reference alleles in the figure legends and methods.

2. In Figure1E, the level of Sax-7 is significantly changed with the addition of inactivated ribozyme, so the text “Sax-7 protein remained almost unchanged” is not very accurate.

Response: We apologize for the confusion and have modified the wording as “while the level of GFP-Rz⁺-fused SAX-7 protein was slightly reduced”.

3. Since the insertion of inactivated ribozyme already leads to reduction in mRNA and protein level of the endogenous genes, the authors should discuss potential reasons. Is that due to a faster degradation rate, or is there some residual activity in the ribozyme? Or other reasons?

Response: We thank the reviewer for pointing this out. The insertion of inactive ribozyme into the 3' UTR of target gene significantly reduced protein expression of several target genes. As suggested by the reviewer, it is possible that the insertion of inactivated ribozyme into 3' UTR may alter mRNA stability, transport and translation. Since the inactive ribozyme is only used as a control in this study, it does not affect the conclusion of this manuscript. Thus, we added the following description to the result section:

“These results suggest that placing the inactive ribozyme in the 3' UTR may affect the expression of some target genes, presumably due to various factors such as defects in mRNA stability, post-transcriptional modification, transport or translation.”

4. In Figure 4, the addition of 400uM Morpholino reversed the gene activation effect. The authors should discuss potential reason of this repression.

Response: We thank the reviewer for this comment. Following the reviewer's suggestion, we tested whether injecting 400 μ M morpholino generally inhibits gene transcription. Thus, we injected 400 μ M morpholino into both *argn-1::gfp* and *argn-1::gfp::inactive ribozyme* strains. We found that the expression levels of *argn-1::gfp* and *argn-1::gfp::inactive ribozyme* were not reduced, suggesting that a high concentration of morpholinos likely does not generally inhibits gene transcription. The cause of this effect is not clear currently.

We have added the following sentence into the result section:

“Notably, the injection of 400 μ M morpholino was ineffective in turning on the expression of endogenous ARGN-1 or rescuing the abnormal mitochondrial morphology. The underlying mechanism behind this observation requires future investigation (Fig. 5e-5g).”

Data 1. Effect of injecting 400 μ M morpholino on the expression of ARGN-1::GFP and ARGN-1::GFP::inactive ribozyme

REVIEWERS' COMMENTS:

Reviewer #1 (Remarks to the Author):

The authors have addressed all my comments/suggestions

Reviewer #2 (Remarks to the Author):

The authors have done a thorough job responding to reviewer comments. I recommend publication.

Reviewer #3 (Remarks to the Author):

The authors have satisfactorily addressed my concerns by adding new data and providing thorough explanations. Their efforts have greatly improved the quality of the manuscript. I believe that the manuscript is now ready for acceptance.